# The Influence of Prebiotics on Wheat Flour, Dough, and Bread Properties; Resistant Starch, Polydextrose, and Inulin

**DOI:** 10.3390/foods11213366

**Published:** 2022-10-26

**Authors:** Fereshteh Ansari, Tatiana Colombo Pimentel, Hadi Pourjafar, Salam A. Ibrahim, Seid Mahdi Jafari

**Affiliations:** 1Razi Vaccine and Serum Research Institute, Agricultural Research, Education and Extension Organization (AREEO), Tehran 148-31975, Iran; 2Research Center for Evidence-Based Medicine, Health Management and Safety Promotion Research Institute, Tabriz University of Medical Sciences, Tabriz 5166-15731, Iran; 3EBM A Joanna Briggs Institute Affiliated Group, Iranian Centre, Tabriz 5166-15731, Iran; 4Federal Institute of Paraná, Paranavaí, Paraná 19044, Brazil; 5Dietary Supplements and Probiotic Research Center, Alborz University of Medical Sciences, Karaj 31497-79453, Iran; 6Food and Nutritional Sciences Program, North Carolina Agricultural and Technical State University, E. Market Street, 1601, Greensboro, NC 24711, USA; 7Nutrition and Bromatology Group, Department of Analytical Chemistry and Food Science, Faculty of Science, Universidade de Vigo, E-32004 Ourense, Spain; 8Department of Food Materials and Process Design Engineering, Gorgan University of Agricultural Sciences and Natural Resources, Gorgan 49138-15739, Iran; 9College of Food Science and Technology, Hebei Agricultural University, Baoding 071001, China

**Keywords:** prebiotics, resistant starch, inulin, polydextrose, farinography-extensography

## Abstract

The addition of prebiotics to bread is one of the most important ways to improve its techno-functional properties. In this study, the effects of resistant starch, polydextrose, and inulin on wheat flour, dough, and bread properties were investigated. The farinography results showed that resistant starch significantly increased the development time (2:18) via a boosting effect; however, polydextrose (1:48) and inulin (1:36) weakened the dough (*p* < 0.05). Inulin, polydextrose, and resistant starch had the greatest effect on reducing water absorption (40, 43.2, and 48.9), respectively, (*p* < 0.05). According to extensography data, the addition of inulin produced the best result in baking compared to other polysaccharides. In terms of baked breads, the samples containing resistant starch had high moisture content that could be due to starch gelatinization and moisture-retention, which delays the staling process of the bread. Inulin, polydextrose, and resistant starch prebiotic ingredients affected the rheological properties of the dough, overall bread quality and organoleptic characteristics; however, resistant starch was the best prebiotic used in this study.

## 1. Introduction

Prebiotics as nondigestible ingredients are digested by the colon’s microflora and produce health-promoting compounds such as various organic acids, short-chain fatty acids, and vitamins [1,2]. The resultant changes in the composition or activity of the intestinal microflora improve the health and well-being of the host by different mechanisms including th following: enhancing satiety and consequent inhibition of obesity, regulating gut movements, preventing diarrhea and constipation, and reducing the deposition of harmful microorganisms, such as Bacillus, Salmonella and Staphylococci in the colon [2,3]. Regular and prolonged use of prebiotics also regulates and stimulates the immune system and increases the body’s resistance to various types of diseases [4,5,6]. Increasing the absorption of vital minerals and vitamins, reducing blood cholesterol and improving insulin sensitivity and positive effects against cancers, especially gastrointestinal cancers and metabolic syndrome, are other beneficial effects of these functional compounds [7,8,9,10].

The European Food Standards Agency (EFSA) has established the dietary reference value for dietary fibers and prebiotics at 25 g per daily diet for adults 18 years or over for the purpose of maintaining normal abdominal functions but acknowledges that higher levels of absorption are more beneficial [11]. The National Health and Nutrition Examination Survey (NHANES) found that U.S. residents 20 years and older use only 61% of their apparent. While, official recommendations regarding the use of prebiotics have not been published, some researchers have suggested 10 g fructo-oligosaccharides (FOS) and 7 g galacto-oligosaccharides (GOS) per day [12,13,14]. The prebiotic effect can be observed at low dosages, such as 2.5–5 g/day for resistant starch [15], 1–6 g/day for inulin [16], and 2–7.5 g/day for polydextrose [17].

Resistant starch is a carbohydrate that resists ingestion in the gastrointestinal tract but ferments in the colon and acts as a prebiotic that feeds the probiotic microorganisms that reside there. Resistant starch is one of the most important prebiotic ingredients that can be used in different kinds of bakery products. For example, enrichment of bread crumbs with resistant starch has been gaining importance and is an impactful approach to enhancing product quality [18,19]. In addition, inulin (fructan) is fermented via the gut microbiome and is considered a prebiotic ingredient. Various studies have indicated that using inulin as a prebiotic improves the nutrition and health benefit properties in different kinds of breads without any adverse impacts on rheological and organoleptic characteristics of the products [20,21,22]. Resistant starch and inulin are examples of potential prebiotics that fulfill this criterion which can also change the farinography and extensography characteristics of dough [18,23,24]. Polydextrose is a type of carbohydrate that is used as a prebiotic and also used as a sweetener and to improve the texture of food products [25,26]. The impacts of this compound have not been extensively studied in bread or other similar products. Therefore, it becomes compelling to study the effects of polydextrose on the rheological and sensory characteristics of bread and to compare it with resistant starch and inulin.

According to experts in this field of study, the type of prebiotics used in products such as pasta or other bakery products that will be heated (>90 °C), should be resistant to high temperatures and should not melt during the preparation process [27]. Moreover, the addition of prebiotics should not result in any undesirable taste and texture in the final products which must also continue to be consumer-friendly in terms of appearance and color [28,29]. One other point that informed our interest in this particular study was the lack of certain essential amino acids such as lysine, threonine and tryptophan, in wheat flour which resulted in a downgrading of the nutritional value of bread. Wheat flour has been enriched by whey that contains a lot of essential amino acids. Adding a small amount of whey can significantly improve the protein quality of wheat flour [30,31,32,33].

In our previous study, we primarily assessed the potential application of prebiotics in baking breads with advanced nutritional characteristics. In that study, we evaluated the size of the particles in the flour, the firmness and moisture content, and sensory properties of the bread during storage. The results were promising and showed that prebiotics not only improved the nutritional content of the product but also augmented the maintenance duration of it [34]. Consequently, we planned to continue with that work by evaluating dough formulated with prebiotics. In the present study, we added whey powder to the flour which has resulted in a slightly different formula. So far, to the best of our knowledge, there have been no studies comparing the effects of these three prebiotic polysaccharides (resistant starch, inulin, and polydextrose) on the farinography and extensography characteristics of the resulting dough. In this study, three different samples of prebiotic flours including inulin and whey powder, polydextrose and whey powder and resistant starch and whey powder were evaluated with regard to their characteristics and the features of the produced dough and bread from these samples. The purpose of this study was thus to investigate which of the prebiotic compounds had the best effect on the physicochemical properties of flour and dough and the physicochemical and sensory properties of baked breads and is acceptable to the food industry.

## 2. Materials and Methods

### 2.1. Formulation of Prebiotic Wheat Flours

Commercially soft white flour was provided from Maragheh Flour Factory (Altin Flour, Maragheh, Iran). In a special mixer (YAYANG, Ziyun, China), the experimental samples were prepared according to Table 1. as 5% by weight of inulin (PYSON CO. LTD., Xi’an, China) + 5% by weight of whey powder (Zarin Laban Pars Company, Tehran, Iran) (In sample), 5% by weight of polydextrose (PYSON CO. Xian, China) + 5% by weight of whey powder (PD sample), and 5% by weight of resistant starch (Hi-maize 260, National Starch, Bridgewater, NJ, USA) + 5% by weight of whey powder (RS sample) each added to the flour mixture and completely mixed and homogenized. The control sample (Blank) was from the same flour with 5% by weight of whey powder but without any prebiotic ingredient addition. Regarding the fact that whey powder was used in all groups, the results were only attributable to prebiotic compounds (In, PD, and RS samples).

### 2.2. Physicochemical Characterization of Flours

The quality of the control and formulated prebiotic flours were tested using American Association of Cereal Chemists (AACC) routine methods such as moisture (AACC, 16), ash (AACC, 08-01), protein (AACC, 46-12), wet and dry gluten, gluten index (AACC, 38-12A), Falling number (AACC 56-81B), and Zeleny test (AACC 56-60) based on the sample’s initial weights [35].

### 2.3. Dough Preparation and Baking Bread

A uniform method for the preparation of dough for bread-making was applied. The basic dough (control dough) was prepared from 2000 g flour consisted of salt (20 g), sugar (40 g), compressed yeast (60 g), and the volume of water required getting to 500 BU of consistency via the farinograph (Section 2.4) [36,37]. Wheat flour was mixed well with all prebiotic treatments according to Table 1. Bread doughs were formulated by blending all elements and fermented for 12 min; then dough portions (100 g) were divided, hand-moulded and sheeted. Doughs were proofed at 32 °C and 85% humidity up to optimal volume growth and baked at 260 °C for 17 min [38,39]. At the end of the baking, the bread was kept at room temperature for some time to cool and then packed in polyethylene bags for sensory testing.

### 2.4. Farinography Tests

All samples of separate doughs (RS, In, and PD samples) were calculated on a flour dry weight basis and were assessed according to the AACC 54-21 method [35]. Resistant starch, polydextrose, and inulin in a dry powder form were first mixed well with the wheat flour into the mixing bowl (300 g) of the farinogragh (YUCEBAS MAKINE, Aliağa, Turkey, with 300 g dish) that was connected with a circulating water pump and a thermostat operating at 30 ± 0.2 °C. The parameters of water absorption percentage, dough development time (DDT), dough stability, mixing tolerance index (MTI), and farinograph quality number (FQN) were determined accordingly.

### 2.5. Extensography Tests

At first, all samples of separate doughs (RS, In, and PD samples), were each prepared in the 300 g mixing bowl of the farinograph. Water and salt were then added to produce the dough samples with a constancy of 500 BU, followed by 5 min of blending. A test quantity (150 g) was formed into a sphere, formed into a cylinder, and clamped into the fermenting cabinet. After 45, 90, and 135 min reaction times in the fermenting cabinet at 32 °C, each dough piece was overextended in the extensograph (YUCEBAS MAKINE, Turkey) device via a hook until rupture according to the AACC 54-10 method [35]. Extensograph equipment gave the extensibility (E), the ratio max., maximum resistance, and energy.

### 2.6. Determination of Baked Bread Properties

To measure the moisture content of baked loaves of bread, the 44-16 AACC method was carried out on day 1 [35]. To determine the bread volume, 2 h after baking, the AACC 10-05 method was applied, and finally, the specific volume of the loaf was determined by the following equation [35]. Tests of crumb firmness were performed according to the AACC method 74-09 [35].
**Specific volume (cm^3^/g) = loaf volume/loaf weight**(1)

### 2.7. Sensory Analysis of Produced Bread

The organoleptic characteristics of final breads prepared from control and prebiotic flours were carried out by a panel of 12 trained people (20–40 years old, non-smokers) at room temperature. The extreme marks for each factor were: appearance 10, color 10, chewiness 15, crust 15, texture 15, aroma 15, and flavor/taste 20 [15].

### 2.8. Statistical Analysis

The sample size was estimated based on the moisture percentage of bread loaves. Assuming the power of 80%, a 95% confidence interval, and an effect size of 2 and the standard deviation of 0.8, the calculated sample size was 3 per group [40].

All tests were performed with three replications. The data were presented using mean (SD). Shapiro–Wilk test carried out to test the normality of data. According to the results, all the variables were normally distributed. Subsequently, three experimental groups were compared using One-Way ANOVA followed by Bonferroni post hoc tests. *p* < 0.05 was considered a significant level. The test was not performed for moisture, Zeleny I, Zeleny II and pH, because the standard deviations of all the groups were 0.

## 3. Results and Discussion

### 3.1. Quality of Formulated Prebiotic Wheat Flours

The results of the physicochemical quality of the control and formulated prebiotic flours are shown in Table 2. The amount of ash in the control sample was significantly lower than other samples (*p* < 0.01) because of the addition of prebiotics. High levels of ash indicate high levels of minerals and the high nutritional value of prebiotic flours. Minerals also play an important role in increasing gluten quality during dough preparation by improving the performance of the gluten network. On the contrary, the moisture content in the control sample was higher than the rest of the samples. Protein content in the tested flours was slightly lower than in the control sample due to gluten dilution by the addition of prebiotics. Furthermore, Zeleny I and II in the control sample was higher than all of the prebiotic samples. Reducing the percentage of humid/wet gluten (H gluten) and dry gluten (D gluten) in samples containing prebiotics compared to the control sample is similar to protein percentage due to the addition of gluten-free materials and the reduction of total gluten content. However, comparing the gluten index among the tested samples, especially in the case of inulin sample, showed that with a significant increase in the quality of gluten and a reduction in gluten content, strengthening the gluten network and improving the performance of flour and dough by storing the gases during the fermentation process have been done. The resulting changes in the Falling number are not significant due to the fact that they are not outside the 200–400 range and are considered to be desirable in all four samples.

### 3.2. Farinograghy Results

The results of farinography tests and analyzing curves are shown in Figure 1. The resistant starch significantly increased the development time and had a boosting effect, but polydextrose and inulin weakened the dough. In the case of stability, all the ingredients added to the flour have played a boosting role in the dough as the dough persists. The amount of loosening of the dough was reduced after 12 min by adding resistant starch and polydextrose (dough reinforcement) and increased with the addition of inulin (weakening of the dough). The amount of loosening of the dough after 20 min was reduced by adding all three ingredients (the effect of dough reinforcement through increased stability). In terms of water absorption, inulin, polydextrose, and resistant starch had the most effect on reducing water absorption, respectively. Considering that inulin had the highest impact on the water absorption percentage, as a result of increasing inulin consumption, the percentage of flour water absorption decreases. This can be attributed to the composition of inulin, i.e., fructose-polysaccharides, which is in accordance with the findings of Mohtarami et al. (2015), Wang et al. (2002), and Peressini and Sensidoni (2009) [30,41,42].

Farinography quality number (FQN) can be considered as a combination of different parameters. According to the results, and as seen in Figure 1, resistant starch, polydextrose, and inulin had a negative effect on the characteristics of the dough in terms of farinography. It is believed that gluten creates specific dough structures and plays a significant role in dough and bread characteristics.

Hence, the dilution of gluten via added fibers causes the breaking of the starch- gluten network and a reduction of dough stability [43]. Consequently, the replacement of a certain portion of wheat flour with some materials like inulin, polydextrose, and resistant starch leads to a decrease in protein content. In contrast, some investigations indicated that particular hydrocolloids and fibers improved rheological characteristics of wheat flour, for instance, water absorption, dough stability, and dough development time [15,44,45]. The dilution of gluten by fibers lonely cannot describe all of the alterations in the addition of fiber to wheat flour. The effect of various kinds of fibers on dough characteristics can be described by reactions between gluten protein of flour and fibers [15,45].

### 3.3. Extensography Results

The results of extensography tests and analyzing curves are shown in Figure 2. Regarding the results of extensography, it can be concluded that the added ingredients (resistant starch, polydextrose, and inulin) to the normal flour played the role of oxidizing or reinforcement in the flour. By increasing the strength of the dough and reducing the extensibility, it increases the ratio number and the energy of the dough. Among the added ingredients according to the energy curve, inulin was predicted to have the best result in baking compared to other samples, but it can happen if we consider the absorption of water in the dough-making process. Probably, inulin due to its physicochemical properties reduces proteolytic degradation and thus improves protein flour behavior. In this regard, Brennan et al. (2004), and Karolini-Skaradzinska et al. (2007) reported the raise in the energy needed for dough transformation by increasing the inulin level [46,47]. It seems that inulin, because of its specific physicochemical properties, reduces proteolytic degradation and thus improves the protein behavior of the flour.

### 3.4. Quality of Prebiotic Baked Breads

The results of moisture content, crumb firmness, and specific volume of cooked breads (two hours after cooking) are shown in Table 3. Among the tested samples, the highest and the lowest moisture content was seen in the loaves containing resistant starch and inulin (RS and In samples) respectively. The moisture content of the loaf contained inulin (In sample) was the least and largely similar to the control sample (Blank). In various studies, crumb firmness was reported with the addition of different hydrocolloids [15,48,49]. In this study, the bread fortified with resistant starch (RS sample) exhibited the highest firmness compared with other samples (In, PD, and, Control samples). The crumb firmness of loaf containing inulin (In sample) was smaller than the control sample. In fact, the level of crumb hardening is affected via water content and it is considered as one of the most significant factors for bread staling.

Various studies have shown that the type and amount of prebiotic compounds seem to affect the rheological characteristics of the dough and so have an impact on the specific volume of the fortified bread loaves [15,50,51]. In this study, the specific volume of the bread fortified with resistant starch (RS sample) was the highest and similar to the sample containing polydextrose (PD sample). Furthermore, the specific volume of the bread fortified with inulin (In sample) was the least and largely similar to the control sample. According to Table 3, it was observed that the effect of resistant starch on the resulting breads was somewhat higher. This sample contains high moisture that could be due to starch gelatinization and moisture-retaining, which delays the staling process of bread. On the other hand, this could be attributed to bread crunching and to some extent a better spread of bread (the ratio of volume to bread weight is greater than the rest of the samples, see Table 3) which also affects sensory properties. This kind of bread spreads in the mouth earlier, so the taste of the food can be better tasted (as if bread is finished earlier than food in the mouth). In a previous different study, the authors investigated the effects of adding prebiotics inulin, resistant starch and polydextrose on the characteristics of wheat flour and breads baked from them. It was concluded that the addition of resistant starch and inulin improved the sensory characteristics of bread and increased their shelf life. Therefore, these two compounds, especially resistant starch, could be used to produce prebiotic breads [34].

### 3.5. Sensory Properties of Final Breads

Sensory features were evaluated by the numerical scoring method. All tests were performed in triplicate and the results were reported as average and standard deviation (Table 4). Based on the results, among the produced bread samples (Figure 3), those containing resistant starch (RS sample) had the highest score in all organoleptic items including total acceptability, with the exception of the appearance and color. On the other hand, breads containing polydextrose (PD sample) had the lowest score for organoleptic indices and total acceptability. In examining the appearance and color of the samples, except for the control sample, in all samples, brown spots can be observed on the bread surface which is an undesirable feature. It seems that these spots are caused by the non-uniform mixing of the added prebiotic compounds into the flour that are nodded in the surface of the dough and burned briefly under the heat. In other words, the reason for the change in the color of the breads during baking, which is caused by the baking temperature and the phenomenon of non-enzymatic browning [33], was uniform everywhere on the surface of the control breads, but on the surfaces of the other tested breads that contained prebiotics, there were scattered brown spots darker, which indicates the occurrence of more intense browning phenomenon.

Because it was an experimental study, the study condition was completely under control and was similar for all the experimental groups rolling out the probability of selection bias. Since all the objective measurements were done with a single standard method and similar devices and personnel, the chance of measurement bias was also very low. The only parameters which were measured subjectively were the organoleptic characteristics. In this case, we kept the participants unaware of the types of samples to reduce the measurement bias.

## 4. Conclusions

In the present study, inulin, polydextrose, and resistant starch prebiotic ingredients positively affected dough rheological properties and bread quality, and organoleptic characteristics. Among all of the studied prebiotic polysaccharides, resistant starch clearly stood out as the superior compound for formulation and production of prebiotic bread. Resistant starch also increased the shelf life and reduced the staling of the bread. According to the results of this article adding prebiotic compounds does not have a significant negative effects on the basic characteristics of bread, however, finding the best prebiotic with the least negative effects (and most positive effects) is valuable and can be used in industrial production.

## Figures and Tables

**Figure 1 foods-11-03366-f001:**
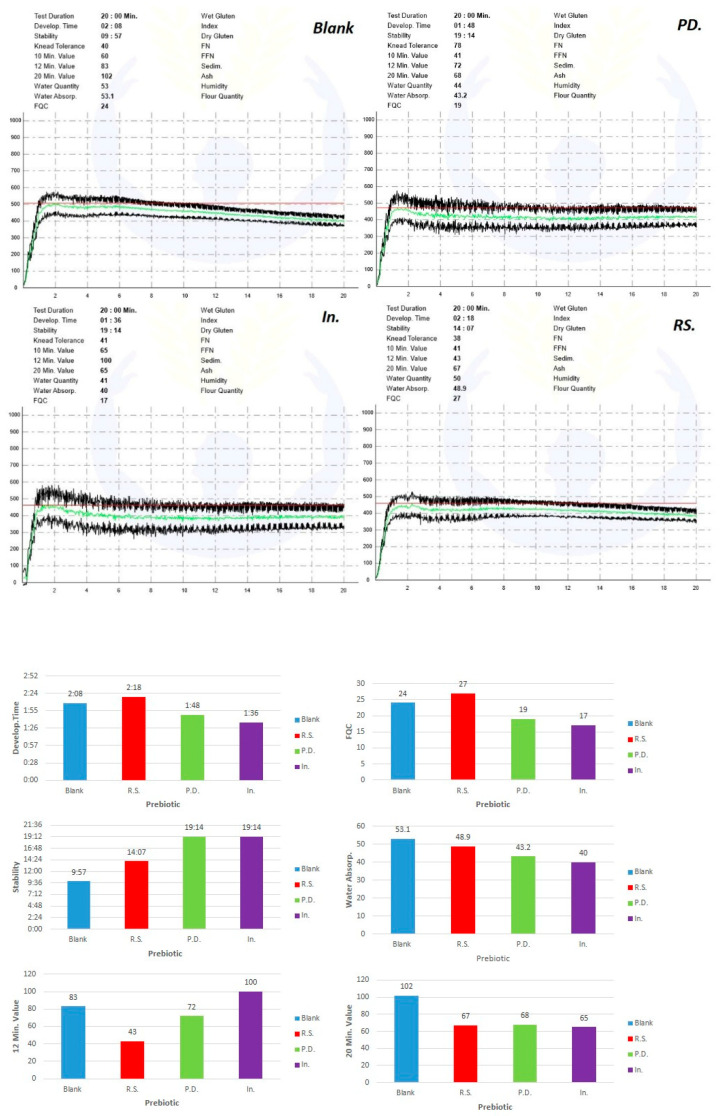
Top subfigure: Original farinograph curves of wheat flour (Blank = control) and wheat flour enriched with polydextrose (PD), inulin (In), and resistant starch (RS); Down subfigure: Analyzing farinograph curves of wheat flour (Blank) and wheat flour enriched with polydextrose (PD), inulin (In), and resistant starch (RS).

**Figure 2 foods-11-03366-f002:**
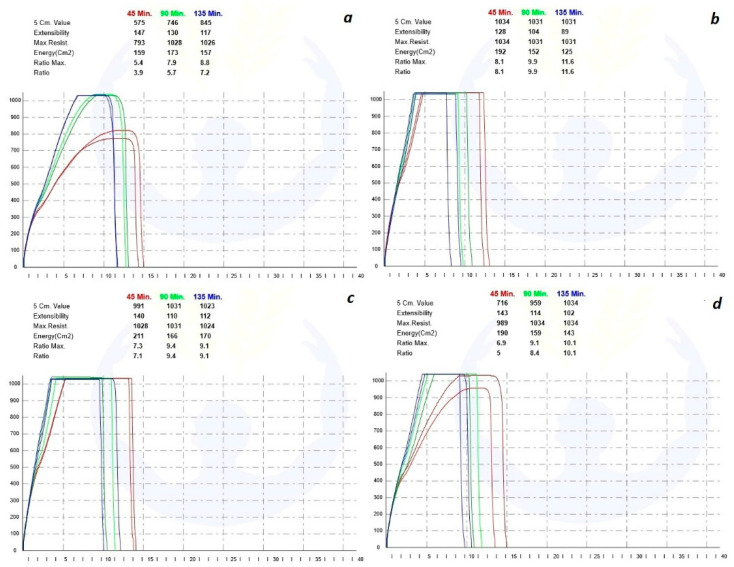
**Top subfigure:** (**a**) Original extensograph curves of wheat flour (Blank = a) and wheat flour enriched with polydextrose (**b**), inulin (**c**), and resistant starch (**d**); **Down subfigure**: Analyzing extensograph curves of wheat flour (Blank) and wheat flour enriched with polydextrose (PD), inulin (In), and resistant starch (RS).

**Figure 3 foods-11-03366-f003:**
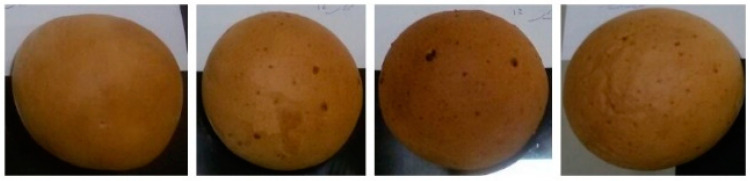
Loaves of control bread and loaves of bread supplemented with inulin (In), polydextrose (PD), and resistant starch (RS). Loaves arranged from left to right Blank/Control, In, PD, and RS samples respectively.

**Table 1 foods-11-03366-t001:** Formulation of prebiotic wheat flours as experimental samples.

Sample	Prebiotic Content (g, %)	Whey Content (g, %)	Flour Content (g, %)	Total Content (g, %)
RS	50, 5	50, 5	1900, 90	2000, 100
PD	50, 5	50, 5	1900, 90	2000, 100
In	50, 5	50, 5	1900, 90	2000, 100
Control	0, 0	0, 0	2000, 100	2000, 100

RS = Samples contain resistant starch, PD = Samples contain polydextrose, In= Samples contain inulin.

**Table 2 foods-11-03366-t002:** Physicochemical characteristics of the dough prepared from different flour samples (Mean ± SD).

Sample	Total Ash (%)	Moisture (%)	Protein (%)	pH	ZelenyI (mL)	ZelenyII (mL)	H Gluten	D Gluten	Gluten Index	Falling Number (s)
PD	0.94	12.9±	9.10±	6.12±	18.00±	21.00±	21±	7.00±	92.85±	322±
	±0.01 ^a^	0.00	0.00 ^a^	0.00	0.00	0.00	0.42 ^a^	0.00 ^ab^	0.49 ^ab^	15.56
In	1.00	13.1±	9.25±	6.21±	21.00±	21.00±	18.5±	6.45±	97.05±	343.00
	±0.07 ^a^	0.00	0.95 ^a^	0.00	0.00	0.00	0.5 ^b^	0.05 ^b^	0.25 ^a^	±8.00
RS	1.04	13.5±	10.25±	6.12±	18.00±	22.00±	24±	7.9±	87.65±	333±
	±0.06 ^a^	0.00	0.21 ^ab^	0.00	0.00	0.00	0.28 ^c^	0.14 ^a^	6.29 ^ab^	1.41
Control	0.70	14.00±	11.10±	6.10±	23.00±	28.00±	26.3±	8.55±	87.3±	354.00
	±0.02 ^b^	0.00	0.42 ^b^	0.00	0.00	0.00	0.28 ^d^	0.35 ^c^	2.83 ^b^	±18.38
***p* Value**	**0.007**	**-**	**0.015**	**-**	**-**	**-**	**<0.001**	**<0.001**	**0.024**	0.242

PD = Samples contain polydextrose, In = Samples contain inulin, RS = Samples contain resistant starch. Significant *p* Values are marked in bold. Different lower-case letters indicate statistically significant differences (*p* < 0.05) between experimental groups. **-= As the SD of the measurements was zero, no statistical analysis was eligible to carry out.**

**Table 3 foods-11-03366-t003:** Moisture content, crumb firmness, and specific volumes of bread loaves (Mean ± SD).

Sample	Moisture Content (%)	Crumb Firmness (N)	Specific Volume (cm^3^/g)
RS	38.30 ± 0.80 ^b^	1.02 ± 0.08 ^d^	3.45 ± 0.42 ^b^
In	34.99 ± 0.11 ^a^	0.45 ± 0.04 ^a^	2.99 ± 0.08 ^a^
PD	36.54 ± 0.18 ^a,b^	0.85 ± 0.09 ^c^	3.25 ± 0.20 ^b^
Control	35.09 ± 0.25 ^a^	0.70 ± 0.02 ^b^	2.94 ± 0.06 ^a^

RS = Samples contain resistant starch, In = Samples contain inulin, PD = Samples contain polydextrose. Same small letters within the same column are not significantly different (*p* > 0.05).

**Table 4 foods-11-03366-t004:** Sensory scores (Mean ± SD) of control bread and loaves of bread supplemented with polydextrose (PD sample), inulin (In sample), and resistant starch (RS sample).

Sample	Appearance	Color	Chewiness	Crust	Texture	Aroma	Taste	Total Acceptability
RS	7.5 ± 0.0	8.5 ± 0.7	14.75 ± 0.3	14.75 ± 0.3	12.5 ± 0.7	15.0 ± 0.0	17.5 ± 0.7	**90.5 ± 0.71**
PD	7.0 ± 0.0	6.5 ± 0.7	13.5 ± 0.7	12.0 ± 1.4	11.5 ± 0.7	11.0 ± 1.4	15.0 ± 0.0	**76.5 ± 3.5**
In	7.7 ± 0.3	9.5 ± 0.7	14.25 ± 0.3	11.0 ± 1.4	12.25 ± 0.3	14.0 ± 0.0	16.5 ± 0.7	**85.25 ± 3.2**
Control	10.0 ± 0.0	10.0 ± 0.0	14.25 ± 0.3	13.0 ± 0.0	12.0 ± 0.0	14.0 ± 0.0	16.0 ± 0.0	**89.25 ± 0.35**

RS = Samples contain resistant starch, PD = Samples contain polydextrose, In = Samples contain inulin.

## Data Availability

The data presented in this study are available on request from the corresponding author. The data are not publicly available due to [The nature of this research, participants of this study did not agree for their data to be shared publicly].

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
