# Peer review of "The Influence of Prebiotics on Wheat Flour, Dough, and Bread Properties; Resistant Starch, Polydextrose, and Inulin"

_foods, 2022, doi:10.3390/foods11213366_

Round 1
Reviewer 1 Report
The title of this study is: The influence of prebiotics on wheat flour, dough, and bread
properties; resistant starch, polydextrose, and inulin. In this study was to evaluate three different samples of prebiotic flours including inulin and whey powder, polydextrose and whey powder and resistant starch and whey powder, regarding their characteristics and the features of the produced dough and bread from these samples.
I commented on the manuscript and the comments are presented below:
Part 1: Introduction.
The Introduction to the study is broad and does not end with a clearly stated p urpose or goals that the Authors wish to pursue. This should be changed. I suggest supplementing the Chapter with additional information related to other new methods and devices in studied research, for example:
“Fiber Preparation from Micronized Oat By-Products: Antioxidant Properties and Interactions between Bioactive Compounds”
“Effect of Supplementation of Flour with Fruit Fiber on the Volatile Compound Profile in Bread”.
“Application of parsley leaf powder as functional ingredient in fortified wheat pasta: nutraceutical, physical and organoleptic characteristics”
Part 2: Material and methods
The Material and methods section provides the reader with enough information to repeat the experiments conducted. The statistical analysis was used to describe the differences.
Part: 3 Results and discussion
For the most part the Results section is well structured.
In the Discussion chapter, there is no full comparison and confrontation with the research of other authors in this area. The results were not fully discussed. A full discussion of the results obtained with other work in this field should be carried out in more aspects. I suggest supplementing the Chapter with additional information, for example:
“Characterization of coloured compounds obtained by enzymatic extraction of bakery products”, “Curcumin and Weight Loss: Does It Work?”, “The use of kidney bean flour with intact cell walls reduces the formation of acrylamide in biscuits”.
Part: 4 Conclusion
The Conclusions chapter contains information obtained after conducting experiments and performing statistical analyzes.
Part: References.
The literature used is appropriate but should be supplementing about the items from the last years of publication about similar problem.
Author Response
Thank you very much for the valuable comments of the respected reviewers. The response to the comments is as follows: (All changes/revisions in the manuscript have been done by Track change)
Reviewer 1:
Open Review
(x) I would not like to sign my review report
( ) I would like to sign my review report
English language and style
( ) Extensive editing of English language and style required
( ) Moderate English changes required
( ) English language and style are fine/minor spell check required
(x) I don't feel qualified to judge about the English language and style
|
Yes |
Can be improved |
Must be improved |
Not applicable |
|
|
Does the introduction provide sufficient background and include all relevant references? |
( ) |
( ) |
(x) |
( ) |
|
Are all the cited references relevant to the research? |
(x) |
( ) |
( ) |
( ) |
|
Is the research design appropriate? |
(x) |
( ) |
( ) |
( ) |
|
Are the methods adequately described? |
(x) |
( ) |
( ) |
( ) |
|
Are the results clearly presented? |
( ) |
(x) |
( ) |
( ) |
|
Are the conclusions supported by the results? |
(x) |
( ) |
( ) |
( ) |
Comments and Suggestions for Authors
The title of this study is: The influence of prebiotics on wheat flour, dough, and bread properties; resistant starch, polydextrose, and inulin. In this study was to evaluate three different samples of prebiotic flours including inulin and whey powder, polydextrose and whey powder and resistant starch and whey powder, regarding their characteristics and the features of the produced dough and bread from these samples. I commented on the manuscript and the comments are presented below:
Part 1: Introduction.
The Introduction to the study is broad and does not end with a clearly stated purpose or goals that the Authors wish to pursue. This should be changed. I suggest supplementing the Chapter with additional information related to other new methods and devices in studied research, for example:
1.“Fiber Preparation from Micronized Oat By-Products: Antioxidant Properties and Interactions between Bioactive Compounds”
2.“Effect of Supplementation of Flour with Fruit Fiber on the Volatile Compound Profile in Bread”.
3.“Application of parsley leaf powder as functional ingredient in fortified wheat pasta: nutraceutical, physical and organoleptic characteristics”
Response: We have changed the purpose of the study at the last sentence of the introduction. Recommended references were cited and used in the text of the manuscript (marked with yellow highlight).
Part 2: Material and methods
The Material and methods section provides the reader with enough information to repeat the experiments conducted. The statistical analysis was used to describe the differences.
Response: Thanks for the positive feedback of reviewer.
Part: 3 Results and discussion
For the most part the Results section is well structured.
In the Discussion chapter, there is no full comparison and confrontation with the research of other authors in this area. The results were not fully discussed. A full discussion of the results obtained with other work in this field should be carried out in more aspects. I suggest supplementing the Chapter with additional information, for example:
- “Characterization of coloured compounds obtained by enzymatic extraction of bakery products”, “Curcumin and Weight Loss: Does It Work?”, “The use of kidney bean flour with intact cell walls reduces the formation of acrylamide in biscuits”.
Response: In the search for similar studies, because few studies were found, the comparison of the results with the data of similar studies was limited. According to the recommendation of the respected reviewer, the comparison of the results with similar studies was revised and the recommended reference was cited and used in the “Discussion” section (marked with yellow highlight) (Please see page 11, all changed have been done by Track change).
Part: 4 Conclusion
The Conclusions chapter contains information obtained after conducting experiments and performing statistical analyzes.
Response: Thanks for the positive feedback of reviewer.
Part: References.
The literature used is appropriate but should be supplementing about the items from the last years of publication about similar problem.
Response: It was tried to use the published and relevant articles of the last few years. Although, in some parts, due to the limited number of related articles, slightly older articles were used. Also, according to the recommendation of the respected reviewer, some relevant and effective articles were cited.
Submission Date
23 August 2022
Date of this review
08 Sep 2022 14:21:01
Reviewer 2 Report
Dear authors,
From my point of view you have some similitude with your other article Effect of inulin, polydextrose or resistant starch on the quality parameters of prebiotic bread. Also in text I didn't understand the firs part of the materials and methods -formulation of prebiotic wheat flours. Please clarified this aspects!
Cordially,
R
Author Response
Thank you very much for the valuable comments of the respected reviewers. The response to the comments is as follows: (All changes/revisions in the manuscript have been done by Track change)
Reviewer 2:
Open Review
( ) I would not like to sign my review report
(x) I would like to sign my review report
English language and style
( ) Extensive editing of English language and style required
(x) Moderate English changes required
( ) English language and style are fine/minor spell check required
( ) I don't feel qualified to judge about the English language and style
|
Yes |
Can be improved |
Must be improved |
Not applicable |
|
|
Does the introduction provide sufficient background and include all relevant references? |
(x) |
( ) |
( ) |
( ) |
|
Are all the cited references relevant to the research? |
(x) |
( ) |
( ) |
( ) |
|
Is the research design appropriate? |
(x) |
( ) |
( ) |
( ) |
|
Are the methods adequately described? |
(x) |
( ) |
( ) |
( ) |
|
Are the results clearly presented? |
(x) |
( ) |
( ) |
( ) |
|
Are the conclusions supported by the results? |
(x) |
( ) |
( ) |
( ) |
Comments and Suggestions for Authors
Dear authors,
From my point of view you have some similitude with your other article Effect of inulin, polydextrose or resistant starch on the quality parameters of prebiotic bread. Also, in text I didn't understand the first part of the materials and methods -formulation of prebiotic wheat flours. Please clarified this aspect!
Response: Thanks for the positive feedback of reviewer. The first part of the materials and methods -formulation of prebiotic wheat flours were clarified in Table 1. For a better understanding of the formulation of different samples in the study, it is recommended to refer to Table 1. According to Table 1, the ingredients required for the formulation of the studied samples were expressed in two forms: weight (grams) and percentage (%).

Reviewer 3 Report
The manuscript with title “The influence of prebiotics on wheat flour, dough, and bread properties; resistant starch, polydextrose, and inulin” was revised.
The effects of resistant starch, polydextrose, and inulin on the wheat flour, dough, and bread properties were investigated by the authors. Relatively to the English I am not a native so I will not evaluate the English of the manuscript.
Good Luck
Author Response
Thank you very much for the valuable comments of the respected reviewers. The response to the comments is as follows: (All changes/revisions in the manuscript have been done by Track change)
Reviewer 3:
Open Review
(x) I would not like to sign my review report
( ) I would like to sign my review report
English language and style
( ) Extensive editing of English language and style required
( ) Moderate English changes required
( ) English language and style are fine/minor spell check required
(x) I don't feel qualified to judge about the English language and style
|
Yes |
Can be improved |
Must be improved |
Not applicable |
|
|
Does the introduction provide sufficient background and include all relevant references? |
(x) |
( ) |
( ) |
( ) |
|
Are all the cited references relevant to the research? |
(x) |
( ) |
( ) |
( ) |
|
Is the research design appropriate? |
(x) |
( ) |
( ) |
( ) |
|
Are the methods adequately described? |
(x) |
( ) |
( ) |
( ) |
|
Are the results clearly presented? |
(x) |
( ) |
( ) |
( ) |
|
Are the conclusions supported by the results? |
(x) |
( ) |
( ) |
( ) |
Comments and Suggestions for Authors
The manuscript with title “The influence of prebiotics on wheat flour, dough, and bread properties; resistant starch, polydextrose, and inulin” was revised.
The effects of resistant starch, polydextrose, and inulin on the wheat flour, dough, and bread properties were investigated by the authors. Relatively to the English I am not a native so I will not evaluate the English of the manuscript.
Response: Thanks for the positive feedback of reviewer.

Round 2
Reviewer 1 Report
The authors referred to the comments from the previous review for the manuscript titled: The influence of prebiotics on wheat flour, dough, and bread properties; resistant starch, polydextrose, and inulin. I accept explanations. They supplemented the discussion with a new literature data strengthens the message and importance of information in the manuscript.
Author Response
Response: Thanks for the positive feedback of reviewer.
Reviewer 2 Report
Dear Authors,
I don't see the comments regarding the other article ....please clarify this aspect with with the editors.
Cordially,
R
Author Response
Response: This article was a continuation of the previous research work (related published article entitled "Effect of inulin, polydextrose or resistant starch on the quality parameters of prebiotic bread"), and they are completely different from each other and their goals and results are different from each other. In the previous article, the characteristics of flour combined with prebiotics and the sensory characteristics of bread baked with those flours were investigated. In the continuation of that, a new research proposal was conducted to complete the previous design work and mainly the Farinography and Extensography characteristics of the resulting dough as well as the quality characteristics of the baked breads in a complementary and different way from the previous research surveyed. Finally, we announce that two different studies have been conducted at two different points in time.